



# Observations of Supermicron-Sized Aerosols Originating from Biomass Burning in South Central Africa

Rose M. Miller[1], Greg M. McFarquhar[2,3], Robert M. Rauber[1], Joseph R. O'Brien[4], Siddhant Gupta[2,3], Michal Segal-Rozenhaimer[5,6], Amie N. Dobracki[7], Arthur J. Sedlacek[8], Sharon P. Burton[9], Steven G. Howell[10], Steffen Freitag[10], Caroline Dang[11]

[1] Department of Atmospheric Science, University of Illinois Champaign-Urbana, Urbana, IL, USA

[2] Cooperative Institute of Mesoscale Meteorological Studies, University of Oklahoma, Norman, OK, USA

[3] School of Meteorology, University of Oklahoma, OK, USA

[4] Department of Atmospheric Science, University of North Dakota, Grand Forks, ND, USA

[5] Bay Area Environmental Research Institute/NASA Ames Research Center, Moffett Field, CA, USA

[6] Department of Geophysics, Porter school of Environmental and Earth Science, Tel-Aviv University, Tel-Aviv, Israel

[7] Department of Atmospheric Sciences, Rosenstiel School of Marine and Atmospheric Science, University of Miami, Miami, FL, USA

[8] Department of Environmental & Climate Sciences, Brookhaven National Laboratory, Upton, NY, USA

[9] NASA Langley Research Center, Hampton, VA, USA

[10] Department of Oceanography, University of Hawaii at Manoa, Honolulu, HI, USA

[11] Universities Space Research Association/NASA Ames Research Center, Moffett Field, CA, USA

*Correspondence to:* Rose M. Miller (rosemm2@illinois.edu)

**Abstract.** During the three years of the ObseRvations of Aerosols above CLouds and their intEractionS (ORACLES) campaign, the NASA Orion P-3 was equipped with a 2D-Stereo (2D-S) probe that imaged particles with maximum dimension (D) ranging from $10 < D < 1280\,\mu m$. The 2D-S recorded supermicron-sized aerosol particles (SAPs) outside of clouds within biomass burning plumes during flights over the Southeast Atlantic off Africa's coast. Numerous SAPs with $10 < D < 1520\,\mu m$ were observed in 2017 and 2018 at altitudes between 1230 m and 3500 m, 1000 km from the coastline mostly between 7-11°S. No SAPs were observed in 2016 as flights were conducted further south and further from the coastline. Number concentrations of black carbon (rBC) measured by a single particle soot photometer ranged from 200 to 1200 $cm^{-3}$ when SAPs were observed. Transmission electron microscopy images of submicron particulates, collected on Holey carbon grid filters, revealed particles with potassium salts, black carbon and organics while energy-dispersive X-ray spectroscopy spectra detected potassium, a tracer for biomass burning, indicating that the submicron particles originated from biomass burning in addition to black carbon. NOAA Hybrid Single-Particle Lagrangian Integrated Trajectory (HYSPLIT) three-day back trajectories show a source in northern Angola for times when large SAPs were observed. Fire Information for Resource Management System Moderate Resolution Imaging Spectroradiometer (MODIS) 6 active fire maps showed extensive biomass burning at these locations. Given the back trajectories, the high number concentrations of rBC, and the presence of elemental tracers indicative of biomass burning, it is hypothesized that the SAPs imaged by the 2D-S are examples of unburned plant material previously seen in biomass burning smoke close to the source.

## 1 Introduction

Global biomass burning (BB) emits on average 2.5 Pg year$^{-1}$ of carbon aerosol, with Africa producing 49% of these global emissions (van Der Werf et al., 2006). Particulates generated by BB scatter and absorb solar radiation, affect the properties and lifetime of clouds (Andreae 1991, Penner et al. 1992, Ackerman et al. 2000, Bond et al. 2013),



and influence regional and global climate (Crutzen & Andreae 1990, Andreae 1991; Bond et al., 2013). Active fire detection from geostationary satellites over central Africa detect BB from woodland, cropland, and grassland fires typically peak in July with ~6 Tg per day of BB combustion (Roberts et al., 2009). Westward transport of aerosols from BB places the plume over an expansive seasonal stratocumulus cloud deck over the southeast Atlantic Ocean (Muhlbauer et al., 2014). The products of BB, including soot aerosol, vary with vegetation type and emission during flaming or smoldering combustion. Previous studies on BB within central Africa observed an abundance of soot aerosols predominately from flaming grass fires (Li et al., 2003).

Soot (or black carbon) aerosol, a byproduct from incomplete combustion during biomass and fossil fuel burning (Bond et al., 2004), still contributes to uncertainty estimates of radiative forcing in global climate models due to its optical property dependence on particle microphysical properties (Bond et al., 2013; Boucher et al., 2013). Combustion of biomass and fossil fuels can produce branching, chain-like (aciniform morphology) soot aggregates from submicron to supermicron sizes (Bond et al., 2004). Following particle generation, soot aggregates can undergo morphological changes that result in a collapse from a fractal structure to a more compact, spherical shape due to their interactions with $H_2O$, $H_2SO_4$ and/or other gaseous species (Zhang et al., 2008). Such physical modification will alter the optical properties of these particles. (Martins et al., 1998, Reid and Hobbs, 1998, Weiss et al., 1992).

In addition, more KCL particles have been found in younger smoke aerosols, whereas more $K_2SO_4$ and $KNO_3$ are present in older smoke aerosol samples. This process happens through reactions from other chemical species present in biomass burning (Li et al., 2003).

Large soot aggregates, defined as $D_p > 1000$ nm, have been observed in field studies of flaming wildfires over the southern Indian Ocean and the Southwestern USA (Chakrabarty et al., 2014). Targeted laboratory experiments have yielded supermicron size soot aggregates ranging from 5 µm to 100 µm (Kearney and Pierce, 2012). This class of soot aggregates have also been observed in the emissions from the Kuwait oil fires in 1991 and were characterized by chain lengths up to 5 µm (Weiss et al., 1992). However, it is unknown whether these supermicron-sized aerosols can be transported long distances from their source regions.

In this study, we report observations of long-range transport of supermicron aerosol particles (SAPs) originating from biomass burning in central Africa using measurements obtained aboard the NASA Orion P-3 Research Aircraft as part of the ObseRvations of Aerosols above CLouds and their intEractionS (ORACLES) field campaign (Redemann et al. 2020; Zuidema et al., 2016). Evidence from transmission electron microscopy (TEM) analysis of aerosol particles, BB aerosol composition analysis, particle shape and size, and prevailing atmospheric conditions together demonstrates that SAPs observed during this campaign were not soot, but rather supermicron-sized unburnt plant material.

## 2 Instrumentation/Data

ORACLES missions were flown in 2016, 2017, and 2018. SAPs were not observed during the 2016 ORACLES field mission, but were on 15 of 24 flights from 2017 and 2018. All of the 2017-18 research flights were based out of the African island nation of São Tomé and Príncipe or Ascension Island in the South Atlantic, whereas the 2016 campaign was based out of Walvis Bay, Namibia.

The likely origin and transport path of the BB plume over the Atlantic Ocean were determined using the location of the observed SAPs during research flights along with the NOAA Hybrid Single-Particle Lagrangian Integrated Trajectory (HYSPLIT) backward trajectory model calculations. Analysis by Wu et al., (2020) suggests that the age of these African BB aerosols to be greater than 7 days. A majority of the BB aerosols sampled during ORACLES were located in the free troposphere (Pistone et al., 2019). Remote sensing analysis of ORACLES has determined there were no systematic difference in aerosol properties found between the air above low-level clouds and above nearby clear sky areas during the daytime (Shinozuka et al., 2020).

## 2.1 ORACLES Field Campaign

ORACLES had three Intensive Observation Periods (IOPs) in successive years to study the atmospheric processes and climate impacts of African BB aerosols (Redemann et al. 2020). A seasonal BB plume is present between July and October where BB aerosols are transported westward from Africa over the southeast Atlantic Ocean. The southeast Atlantic hosts one of the three permanent subtropical stratocumulus cloud decks in the world. Understanding the impact and interactions between this aerosol plume and clouds has been emphasized in IPCC



reports (Myhre, 2013). Aerosol-cloud interaction is one of the largest uncertainties in estimates of future climate from climate models. One of the objectives of ORACLES was to evaluate the interaction of these BB aerosols with stratocumulus clouds and determine their possible direct and indirect radiative effects.

The 2016 IOP was based out of Walvis Bay, Namibia. No SAPS were detected during the 2016 IOP. The 2017 and 2018 IOPs were based out of São Tomé. In these IOPs, 26 successful research flights were carried out in the region between 0°-15°S and 12°E- 15°W, further north than the 2016 flight domain. Instrumentation to detect and
measure aerosol properties was included in the payload of the P-3 aircraft all three years.

## 2.2 2D-S Stereo Probe

Two-dimensional shadowgraph images with 10 µm pixel resolution were collected in situ from the 2D-S
optical array cloud probe (Lawson et al., 2006) about 800-1200 km west of Angola as part of all three ORACLES field campaigns. Seven out of thirteen research flights between 12 August and 31 August 2017 detected SAPs within flight legs sampling biomass plumes above the stratocumulus cloud deck between 1230 m – 3500 m MSL. In 2018, nine out of thirteen research flights between 27 September and 19 October 2018 detected SAPs. Particle images were processed using the University of Illinois/University of Oklahoma optical array processing program (UIOOPS)
(McFarquhar et al., 2018) and the Airborne Data Processing and Analysis (ADPAA) package (Delene, 2011). The 2D-S is normally used to capture cloud and ice particle images, but in this study images corresponding to SAPs were collected in the biomass aerosol plumes. SAPs were identified with 2D-S imagery only during periods when the aircraft was not in cloud, which were indicated by periods when the King hot wire probe liquid water content <0.01 g m $^{-3}$. The absence of clouds in all areas where SAPs were observed was confirmed by examination of the forward
video on the aircraft.

## 2.3 Cloud and Aerosol Spectrometer/Cloud Droplet Probe

The cloud and aerosol spectrometer (CAS) measures smaller aerosol and cloud hydrometeor size distributions for $0.51 < D < 50$ µm and relies on light-scattering rather than imaging techniques (Baumgardner et al., 2001). Data were processed using the Airborne Data Processing and Analysis (ADPAA) software package (Delene,
2011). In 2018 there was an instrument malfunction and the data were not usable for this research. However, data from a Cloud Droplet Probe (CDP) are available for 2018. Like the CAS, the CDP measures size distributions from the forward scattering of light from target particles. It sizes particles between 2 and 50 µm.

## 2.4 Aerosol Mass Spectrometer

The Aerodyne Time-of-Flight Aerosol Mass Spectrometer (ToF-AMS) Operated by the Hawaii Group for
Environmental Aerosol Research (HiGEAR) was used to determine non-refractory submicron aerosol composition within the BB plume by impaction of aerosols on a vaporizer at 650°C (Howell et al., 2014; Shank et al., 2012; DeCarlo et al., 2006; Jayne et al., 2000). This was followed by electron ionization and time-of-flight mass spectral analysis. Size-resolved composition was quantified by measuring the arrival times of the aerosol at the vaporizer, as in Drewnick et al., (2005). ToF AMS data from the 2017 and 2018 campaigns were used to determine quantitative
aerosol composition within the BB plume during the times that SAPs were observed by the 2D-S. The AMS inlet had a cutoff at approximately 700 nm.

## 2.5 Single Particle Soot Spectrometer

During the ORACLES campaigns a single particle soot spectrometer (SP2; Droplet Measurement Technologies, Revision D) was used to detect refractory black carbon (rBC) aerosol particles. The SP2 detects
individual rBC particles through laser-induced incandescence (Schwarz et al., 2006; Moteki and Kondo, 2007). The incandescence signal can probe particles with mass-equivalent diameter of rBC from nominally 80–500 nm assuming a rBC density of 1.8 g cm$^{-3}$. The SP2 was calibrated with fullerene soot (Alfa Aesar; stock #40971. lot#: F12S011). Soot is also commonly referred to as black carbon (BC), elemental carbon (EC), light absorbing carbon (LAC), and refractory black carbon (rBC) (Buseck et al., 2012; Petzold et al., 2013). In the present paper all these particle types
are assumed to be equivalent and the term soot is used to describe them except for the black carbon noted on the aerosol filters.

## 2.6 Aerosol Filter System





An in situ aerosol filter system (AFS) obtained aerosol filters which were then analyzed to provide the chemical composition of aerosols within the BB plume for the 2017 and 2018 campaigns. Particles collected on the filters were analyzed using transmission electron microscope (TEM) and energy dispersive x-ray spectroscopy (EDS) to determine morphology and composition following the techniques of past research for aerosols sized between 30 nm and 5 µm (Echalar et al., 1995, Gao et al., 2003). The inlet size for the AFS allowed aerosols of 2.0 - 3.1 µm diameters, with a 50% collection efficiency depending on the altitude, to be sampled. Filter samples were only collected within the BB plume. For 2017 only two filters were collected and analyzed with the above techniques as the other filters could not be analyzed because of technical problems.

**2.7 P-3 Aerosol Inlet**

The University of Hawaii's shrouded diffuser inlet is thoroughly described in (McNaughton et al., 2007) and was tested during the NASA DC-8 Inlet Characterization Experiment. The volumetric flow rate was proportional to airspeed, was maintained within 5% of the isokinetic flow rate, and had a shrouded constant-area flow region around the inlet with a 4.5-degree diffuser half-angle and a 3.8 cm inner diameter tube with the largest possible radius of curvature to complete a 45 degree bend to bring the air into the fuselage. McNaughton et al., (2007) found that the 50% transmission efficiency of the inlet is 3.1 µm geometric equivalent diameter (with a particle density of 2.6 g cm$^{-3}$) at the surface and 2.0 µm at 12 km. These cut offs will be used for the SP2, AMS, and AFS.

**2.8 NOAA HYSPLIT**

NOAA's HYSPLIT model, January 2017 revision (854) version (Draxler and Hess 1998, Stein et al., 2015) was used to calculate air parcel backward trajectories to determine air mass source regions during the 2017 and 2018 campaigns. The HYSPLIT model was initialized with the Global Data Assimilation System (GDAS) at a 0.5˚ grid spacing. Backward trajectories were initiated at the altitude and location where the SAPs were observed; each backward trajectory was run until the air parcel was within 500 m of the surface, times which spanned between 48 and 128 hours. The trajectories were used to determine the possible location of air parcels and establish source relationships between the BB plume and the observed SAPs. Since air parcels may be lofted through the boundary layer by the heat of fires, which is not accounted for in HYSPLIT, using the 500 m above the surface criteria as the back trajectory endpoints represents the easternmost locations where the parcels likely originated over Africa.

**2.9 CALIPSO and HSRL**

The Cloud-Aerosol Lidar and Infrared Pathfinder Satellite Observation (CALIPSO) satellite provided the location of clouds and atmospheric aerosol loading over the southeast Atlantic region. CALIPSO combines an active lidar instrument with passive infrared and visible imagers to probe the vertical structure and properties of thin clouds and aerosols over the globe (Winker et al., 2003). For this research CALIPSO data from an overpass on August 30$^{th}$, 2017 was used to show the presence of the low-level aerosol airmass being advected off the coast of central-west Africa. On board the P-3 research aircraft for the 2017 and 2018 campaigns there was a similar but more capable active lidar instrument, the High Spectral Resolution Lidar (HSRL-2), that can discriminate between aerosol and molecular signals to measure aerosol extinction, backscatter and depolarization, and clouds (Eloranta et al., 2008). The lidar instruments permitted characterization of the spatial and vertical distributions of the aerosols (Burton et al., 2018, Sawamura et al., 2017).

The HSRL-2 and Cloud-Aerosol Lidar with Orthogonal Polarization (CALIOP) lidar were used to identify the location of a large aerosol plume over the southeast Atlantic Ocean. A large aerosol plume situated 1.5-4.0 km above the stratocumulus cloud deck was frequently observed both by the CALIOP in 2017 and HSRL-2 Lidar in both 2017 and 2018. In situ sampling of this aerosol plume was carried out by the P-3 research aircraft for 24 flights, with 15 flights measuring SAPs.

**2.10 MODIS, FIRMS**

Satellite remote sensing instrumentation was used to determine source origins for SAPs. The Fire Information of Resource Management System (FIRMS) uses near real-time Moderate Resolution Imaging Spectroradiometer (MODIS) data to estimate thermal anomalies and fire locations. The MODIS Collection 6 was processed by NASA's Land, Atmosphere Near real-time Capability for Earth Observing System and Land (EOS), Atmosphere Near real-time Capability for EOS (LANCE) using swath products (MOD14/MYD14). The thermal anomalies and active fires represent the center of a 1 km pixel that is flagged by the MODIS fire and thermal anomalies algorithm (Giglio et al., 2003) as containing one or more fires within the pixel.



## 3 Case Studies

Case studies for two pairs of flights are presented – one pair conducted on 30 August and 31 August from the 2017 ORACLES campaign, and a second pair conducted on 3 October and 4 October from the 2018 ORACLES campaign. These flights were chosen as case studies because flights on the consecutive days provided the opportunity to sample the same plume and allow examination of how the BB plume located over the southeast Atlantic Ocean evolved over time, and whether this evolution influenced the way that the SAPs changed and aged.

### 3.1 30– 31 August 2017

Two sequential research flights, RF 11 and RF 12, occurred on 30 and 31 August. 700 mb (~3000 m), 850 mb (~1500 m) and surface maps for 30 August at 1200 UTC are shown in Fig. 1 a-c.  The wind fields in Figs. 1 a-c together show that the BB plume was only transported westward over the stratocumulus deck at higher levels after the plume was lofted to around 3000 m.

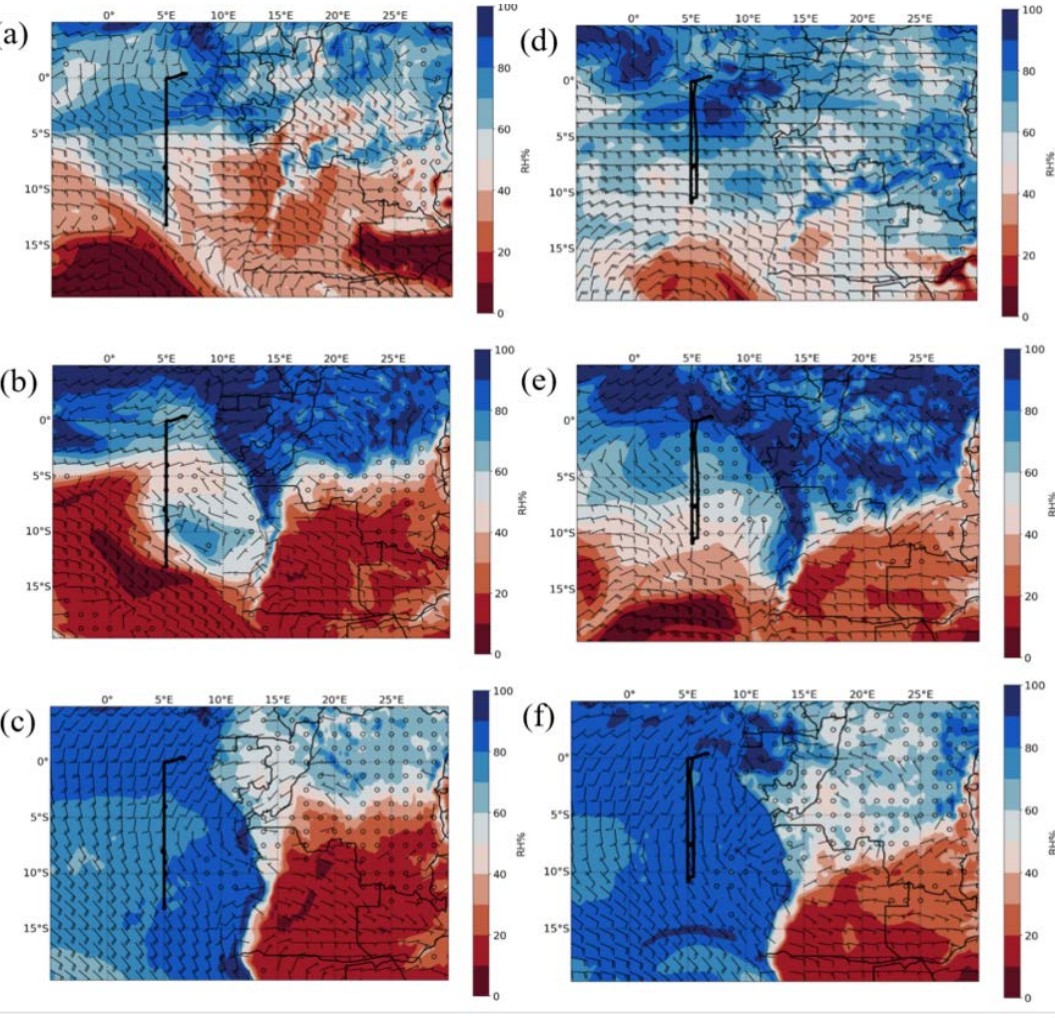

**Figure 1:** European Centre for Medium-Range Weather Forecasts 0-hour reanalysis at 12:00 UTC on August 30[th], 2017 (a-c) and October 2[nd], 2018 (d-f). (a,d) 700 mb relative humidity, along with wind (b,e) 850 mb relative humidity, and wind, (c,f) Surface relative humidity and wind. Wind barbs are as follows: small barbs are 5 m s[-1] and large barbs are 10 m s[-1].  Flight track is shown in black line.





In RF 11, 71 SAPs were measured ranging in size between 10-250 µm along their largest dimension D (Fig. 2). During RF 11 the plume was sampled at 3500 m altitude at 5ºE, and between 4°-10°S.  In contrast, in RF 12, about 24 hrs later, 12 SAPs with D between 800-1520 µm were detected at 2500 m at 2.2ºW, and 5°S, further west of Angola, as denoted by the red stars in Fig. 3.

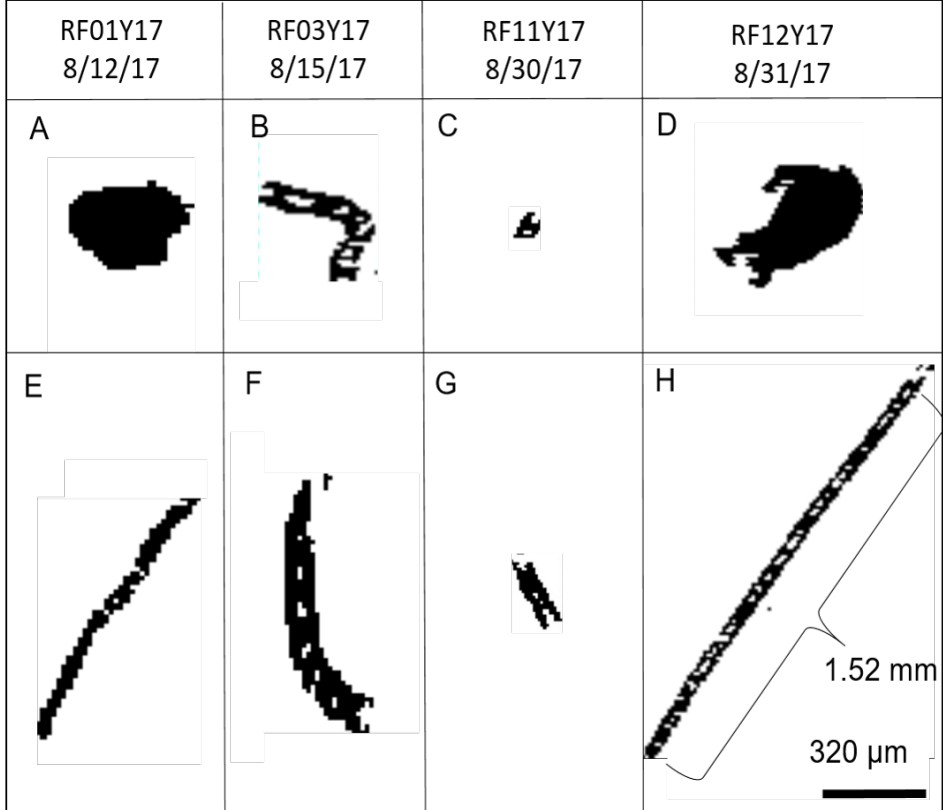

**Figure 2.** Examples of SAPs recorded within the biomass burning plume by the 2D-S probe during the ORACLES 2017 campaign. SAPs were observed to have a wide range of size and shapes ranging from 10-1520 µm in length.


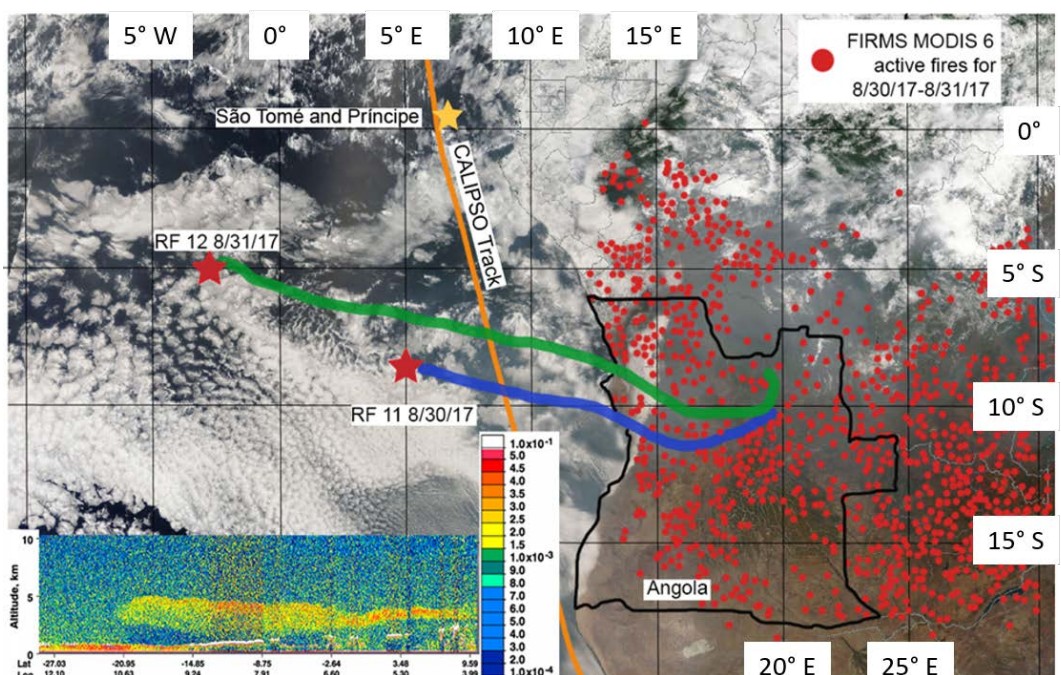

**Figure 3.** NOAA HYSPLIT backward trajectories from RF 11 and RF 12 overlaid with visible imagery of south central Africa from the Moderate Resolution Imaging Spectroradiometer (MODIS) sensor aboard the Terra satellite at 9:40 UTC for August 30th -31st, 2017 (*Image obtained from NASA Near Real-time (NRT) data archive*). Fire Information of Resource Management System (FIRMS) MODIS 6 active fire hot spots for August 30th-31st, 2017 (red dots). Inset: 532 nm attenuated backscatter return signal from the Cloud-Aerosol Lidar with Orthogonal Polarization (CALIOP) lidar aboard the Cloud-Aerosol Lidar and Infrared Pathfinder Satellite Observation (CALIPSO) satellite showing the vertical distribution of aerosols on August 30th, 2017 (*Image obtained from NASA CALIPSO data archive*). The color scale indicates the strength of the lidar return signal, 532 nm total attenuated backscatter, km$^{-1}$ sr$^{-1}$: boundary layer clouds tops as white; aerosols as green, yellow, and red.

During the flights the biomass burning plume layers were identified from HSRL imagery. Selected layers were then sampled by in situ probes as the P-3 flew through the plume. At each time that SAPs were detected with the 2D-S imagery on these flights, the Cloud and Aerosol Spectrometer (CAS) observed aerosols with D ranging from 0.51 µm to 50 µm. The CAS size distributions in Fig. 4 during these time periods represent aerosol distributions outside of cloud, as confirmed by examination of the forward video camera. During RF 11 about 100 more particles





were measured in the larger bin sizes between 10-50 µm compared to RF 12. The CAS did not report any aerosols or
cloud droplets outside of aerosol plumes with SAPs and clouds.

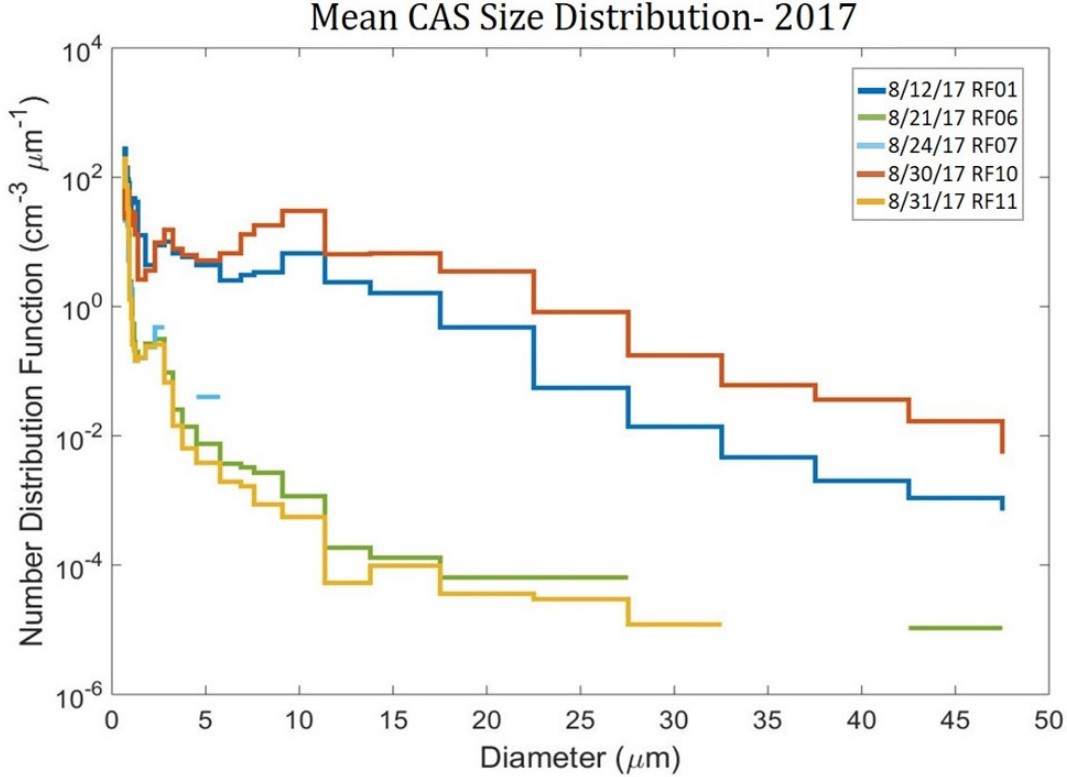

**Figure 4.** Mean cloud and aerosol spectrometer (CAS) aerosol size distribution for five research flights during a five
minute interval when 2D-S SAPs were measured within the BB plume.

From the location of the SAPs, HYSPLIT 48 hr backward trajectories were calculated to estimate the origin
of the air parcels containing the SAPs. Trajectories from both locations showed that SAPs measured at an altitude of
3500 m (RF 11) and 2500 m (RF 12) had their respective air parcels passing over Northern Angola about two and
three days earlier, respectively (Fig. 3). The recorded FIRMS active fires from August 30-31 shows a large number
of active fires throughout Angola and central Africa, and therefore a large source of biomass smoke entering the
atmosphere (Fig. 5). On 30 August a CALIPSO overpass captured a large aerosol plume between 1.5 km and 4.0 km
above sea level transported westward over the Atlantic Ocean and over the stratocumulus cloud deck from the many
fires from Angola and central Africa (Fig. 3).

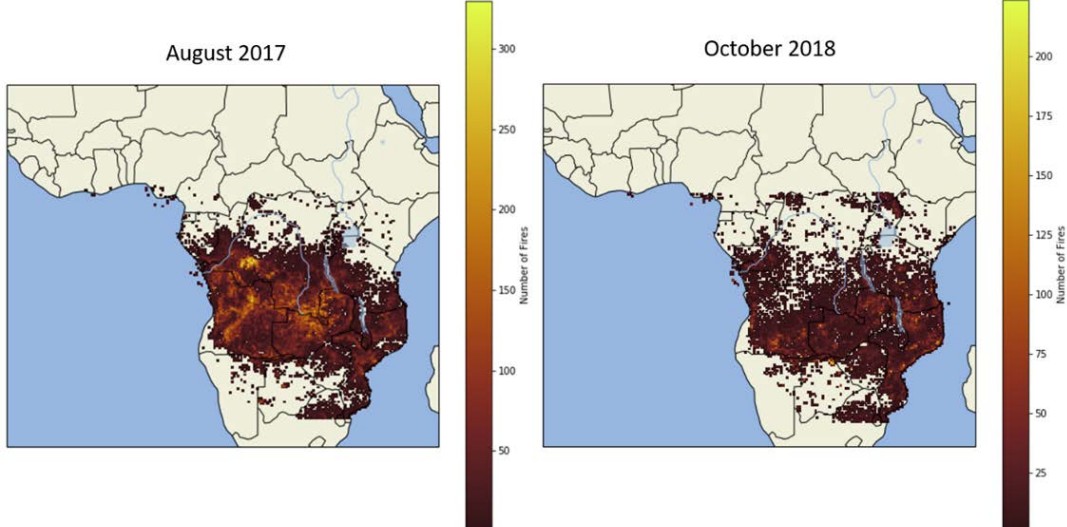

**Figure 5.** FIRMS MODIS 6 active fire map data for the month of August 2017 and October 2018 showing extensive biomass burning in central Africa.

  The altitude and location within the BB plume that the SAPs were observed for RF 12 occurred at 2500 m and 820 km west of Angola. Larger diameter SAPs (D >50 µm) were observed on the 2D-S during the RF 12 flight. These would have not been detected by the CAS. The particles found on RF 12 most likely were either unburned plant material or supermicron black carbon aggregates that were formed near the fire and transported in the BB plume over the Atlantic Ocean.

  Filter samples were acquired by the AFS during the same time that the 2D-S observed SAPs during RF 11 and RF 12 (Fig. 6). The filter samples from both RF 11 and RF 12 contained commonly observed accumulation mode soot aggregates, organic matter, and dust, all with D < 3.1 µm because of the inlet cutoff, significantly smaller than the particles measured by the 2D-S. Energy dispersive X-ray spectroscopy (EDS) was performed on aerosol deposited on the filters (Fig. 7). The soot particles showed silicon inclusions and the organic particles contained potassium, both molecular markers for biomass burning emissions (Andreae et al. 1998). The filters contained numerous black carbon
and organic particles that were captured during the time that the 2D-S observed SAPs. It is therefore likely that the SAPs are unburnt plant material.





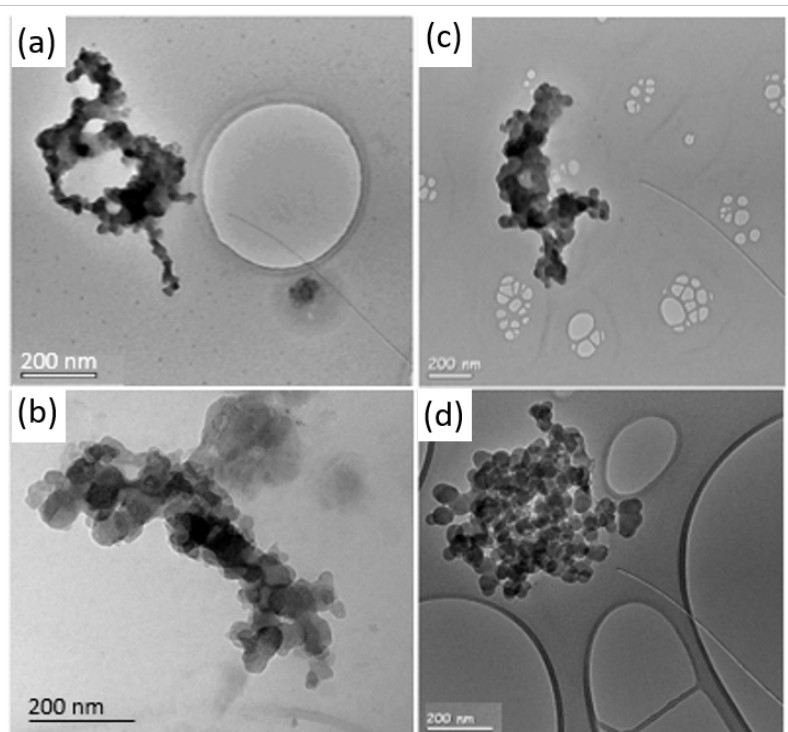

**Figure 6.** Transmission electron microscope/microscopy (TEM) images of black carbon particle aggregates collected on August 30[th], 2017 (A), August 31[st], 2017 (B), October 2[nd], 2018 (C), and October 3[rd], 2018 (D) on gold grid polycarbonate filters.






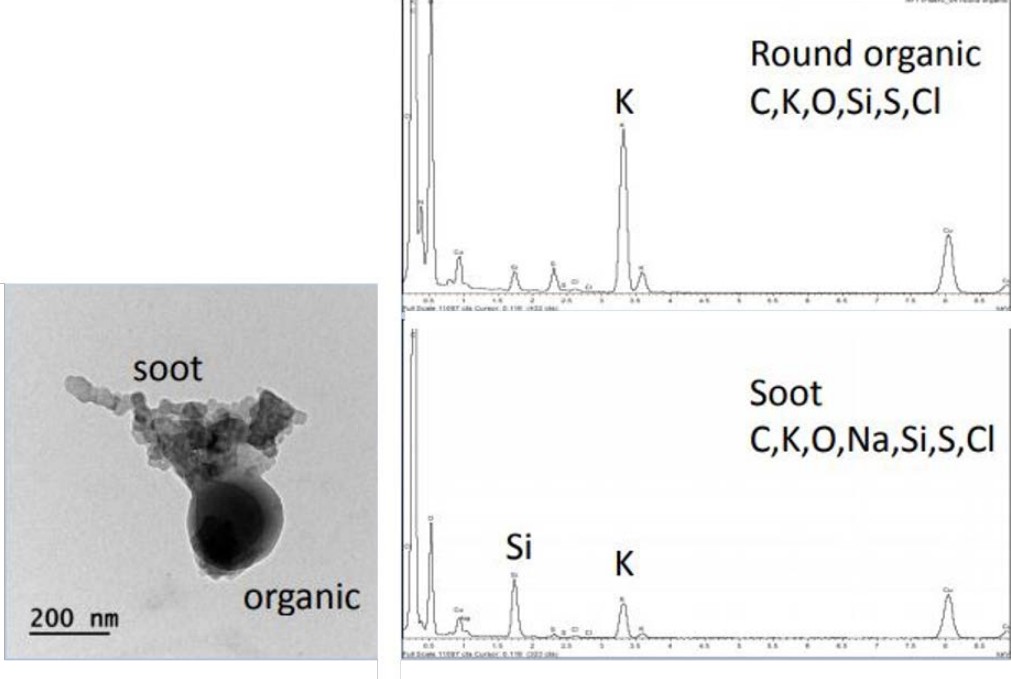

**Figure 7.** Energy-Dispersive X-ray Spectroscopy (EDS) elemental analysis of an organic/soot particle from RF 11, August 30th, 2017.

### 3.2 2 October – 3 October 2018

The RF 03 and RF 04 case studies presented here occurred on 2 and 3 October 2018, and like the 2017 case studies, were two sequential flights where SAPs were observed. The wind fields at the surface, 850 mb, and 700 mb were similar to 2017 in that the BB plume was only transported westward over the stratocumulus deck at the higher altitude (Fig. 1d-f). RF 03 sampled 102 SAPs ranging from 10 <D < 350 µm, and three large (700-1000 µm) SAPs, while RF 4 sampled 48 SAPs with 600 <D < 1000 µm, which were mostly linear SAPs (Fig. 8).




| RF01Y18<br>9/27/2018 | RF02Y18<br>9/30/2018 | RF03Y18<br>10/02/2018 | RF04Y18<br>10/03/2018 |
|---|---|---|---|
|  |  |  |  |
|  |  |  | 640 µm |

**Figure 8.** SAPs recorded within the BB plume by the 2D-S probe during the ORACLES 2018 campaign. SAPs were observed to be a variety of size and shapes ranging from 10-1000 µm in length.

The FIRMS MODIS 6 active fire map data for the 2 and 3 October 2018 showed fewer active fires compared to 2017, all shifted further south due to the start of the rainy season in central Africa (Fig. 5b). RF 03 and RF 04 in 2018 occurred in a similar sampling region (centered at 5°E and 7°S) as opposed to RF 11 and RF 12 during 2017 that sampled different areas. Backward air parcel trajectories from the location of the SAPs (red stars) placed the source region in central Angola for RF 04 and eastern Angola for RF 03 (Fig. 9). The production of SAPs could be correlated with fire intensity as very intense fires can inject copious amount of material into the atmosphere as seen in SAFARI-92 (Canut et al., 1996). The SAPs observed in RF 03 had smaller D overall compared to RF 04 which recorded the largest particle length of 1000 µm. The larger particles of RF 04 were also measured at an altitude of 1300 m, about 1200 m lower in elevation than the particles observed in RF 03 at 2500 m. Larger particles were more often sampled at the bottom layer of the BB plumes due to gravitational settling

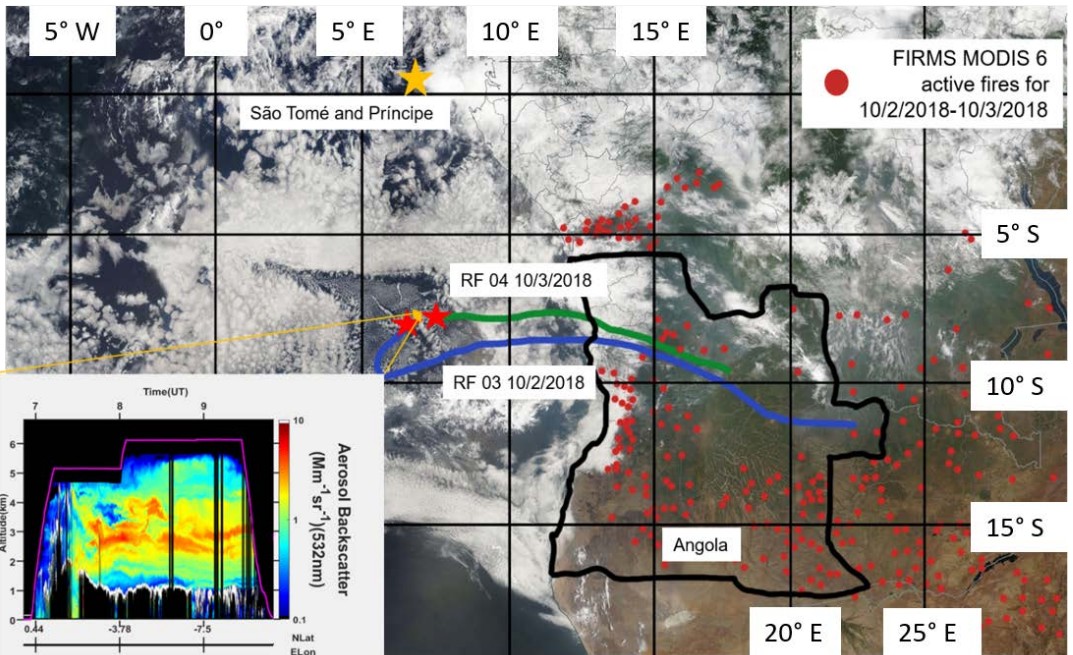

**Figure 9.** Ensemble-averaged NOAA HYSPLIT backward trajectories from RF 03 and RF 04 overlaid with visible imagery of south central Africa from the MODIS sensor aboard the Terra satellite for October 2nd-3rd, 2018 (*Image obtained from NASA Near Real-time (NRT) data archive*). FIRMS MODIS 6 active fire hot spots for October 2nd-3rd, 2018 (red dots). Inset: 532 nm backscatter return signal from the HSRL lidar aboard the P-3 research aircraft showing the vertical distribution of aerosols on October 3rd, 2018. The color scale indicates the aerosol backscattering coefficient: boundary layer cloud tops as white; aerosols as green, yellow, and red (indicating low, medium, and high loads respectively).

## 4 Discussion of 2017-18 observations

The locations and conditions where SAPs were measured varied widely during flights, but when observed were always observed within the BB aerosol plume throughout the 2017 and 2018 campaigns (Fig. 10). In 2017 the sampling area was between 5°S-9.5°S and 14°W-5°E. Research flights conducted in 2018 covered a larger area compared to 2017 with a latitude and longitude range from 1°S-13°S and 5°E-10.5°E, sampling about 420 km closer to the coast of Angola. The altitude of the in-plume sampling ranged between 1230-4000 m for 2017 with SAPs measured lower in the plume between 1230-3500 m. The 2018 plume sampling ranged between 1300-2500 m with SAPs measured throughout that range. The differences in locations sampled between 2017 and 2018 may have impacted the number and size of SAPs recorded and the chemical composition of the overall BB plume.

The 2017 flights took place in August, as the biomass burning season ramped up within central Africa. The 2018 flights were conducted in the month of October, which is the end of the biomass burning season. In 2018, SAPs were observed along the same 5°E longitudinal line as during 2017, but were found over a larger latitudinal range compared to 2017, namely between 3°S and 15°S, compared to between 8°S and 9°S for 2017. In 2018, flights were conducted closer to the coast of Angola than in 2017, but no SAPs were detected by the 2D-S probe closer to the coast; they were only observed further west around 5°E (Fig. 10). The sampling was conducted at altitudes lower than 2000 m closer to the coast. From Fig. 1, the winds at these lower levels were less likely to transport the BB plume to regions sampled by the aircraft. During 2018 the fires in central Africa were less numerous as this was near the end of the BB season in the region. More SAPs were measured overall in 2018 than 2017. This could have resulted from an increase in fire intensity. Higher intensity fires with active flaming with high combustion temperatures most likely produce more SAPs than smaller smoldering fires as higher intensity fires inject more aerosols and unburned plant material into the atmosphere.

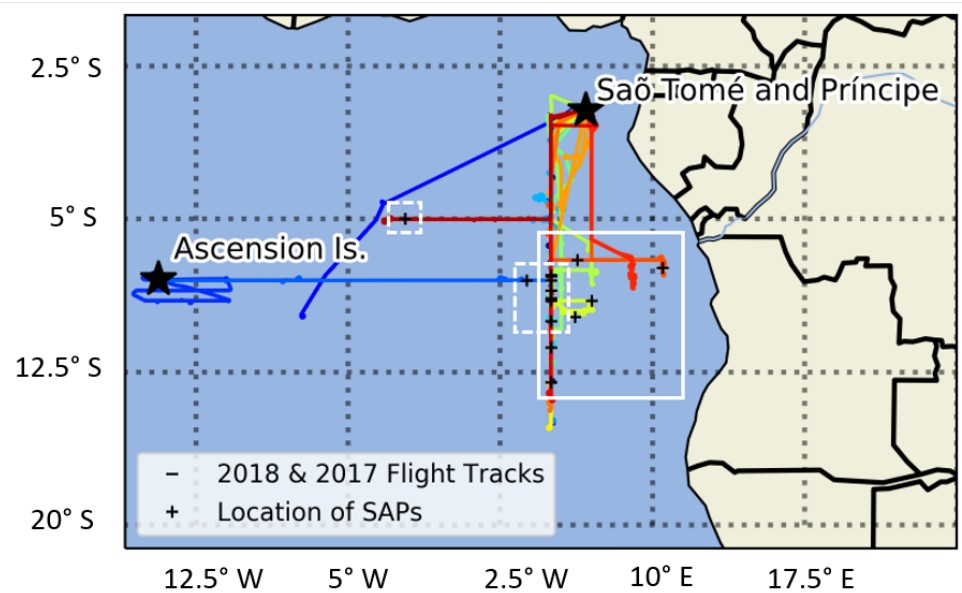

**Figure 10.** Research fight tracks from the 2017 and 2018 Intensive Observation Periods (IOPs) of ORACLES based out of São Tomé and Príncipe over the southeast Atlantic Ocean. The 2017 and 2018 IOPs were off the coast of Angola. A few flights were conducted westward to Ascension Island for clear air sampling. Locations of observations of SAPs are denoted by (+). Area of observed SAPs in 2017 are denoted by the dashed white line, and in 2018 by the solid white line.


Tables 1 and 2 summarize the data, and the range of altitudes, temperatures, and locations at which the SAPs were measured, along with the number of SAPs, the chemical composition of the plume in which they were embedded, the HYSPLIT source areas, and the plume age estimated from HYSPLIT trajectories and WRF model runs described in Mallet et al. (2019). The chemical composition of the aerosol plume within an averaged five-minute time span varied widely on days where SAPs were or were not observed as seen in Tables 1 and 2. For example in Table 1 days


that did not record SAPs had concentrations of organics between 2.1 - 25 µg m$^{-3}$ compared to days where SAPs were measured that had concentrations from 18 - 50 µg m$^{-3}$. Aerosol plumes where SAPs were observed had an overall larger concentration spread of organics, ammonium, sulfates, nitrates, and soot which would be consistent with more intense fires producing more emissions.



**Table 1.** 2017 research flights conducted between 8/12/2017 – 8/31/2017. All concentrations and temperature values are taken at a 5 minute average when supermicron-sized aerosol particles (SAPs) are observed within the BB plume. Some data values were not available and were left blank. Soot number and mass concentrations were obtained from the Single Particle Soot Photometer (SP2). Organics, nitrates, sulfates, and ammonium were obtained from the Aerosol Mass Spectrometer (AMS). Plume age was determined from Hybrid Single-Particle Lagrangian Integrated Trajectory HYSPLIT and Weather Research and Forecasting (WRF) models.

| Date | Flight | Latitude | Longitude | Altitude (m) | Temperature (C) | # of SAPs | HYSPLIT Source Region | Plume Age (Days) |
|---|---|---|---|---|---|---|---|---|
| 8/12/2017 | 17RF01 | 8.5°S | 5°E | 1230 | 22.4 | 15 | Northern Angola | 3 |
| 8/13/2017 | 17RF02 | 9°S | 4°E | 2700 | 12.5 | 0 | Far East Angola | 2.5 |
| 8/15/2017 | 17RF03 | 8.9°S | 5°E | 2100 | 20.0 | 55 | Northern Angola | 2 |
| 8/17/2017 | 17RF04 | 8°S | 6°W | 1800 | 16.2 | 0 | Gabon | 5 |
| 8/18/2017 | 17RF05 | 8°S | 14°W | 1800 | 13.3 | 0 | South America | 8 |
| 8/21/2017 | 17RF06 | 8°S | 3.8°E | 2900 | 16.8 | 9 | Central Angola | 2 |
| 8/24/2017 | 17RF07 | 8°S | 5°E | 2100 | 17.5 | 3 | Northern Angola | 2 |
| 8/26/2017 | 17RF08 | 5°S | 5°E | 4000 | 13.2 | 0 | Central Congo | 3 |
| 8/28/2017 | 17RF09 | 9.5°S | 5°E | 4000 | 8.0 | 0 | Congo | 3 |
| 8/30/2017 | 17RF10 | 9°S | 5°E | 3500 | 9.8 | 71 | Northern Angola | 3 |
| 8/31/2017 | 17RF11 | 5°S | 2.2°W | 2500 | 15.4 | 12 | Central Angola | 2 |

| Date | Flight | Organics Concentration ($\mu g\ m^{-3}$) | Nitrate Concentration ($\mu g\ m^{-3}$) | Sulfate Concentration ($\mu g\ m^{-3}$) | Ammonium Concentration ($\mu g\ m^{-3}$) | Soot Number Concentration (# $cm^{-3}$) | Soot Mass Concentration ($\mu g\ m^{-3}$) |
|---|---|---|---|---|---|---|---|
| 8/12/2017 | 17RF01 | | | | | 780 | 2.1 |
| 8/13/2017 | 17RF02 | 12 | 3.1 | 4.2 | 1.8 | 650 | 1.75 |
| 8/15/2017 | 17RF03 | 38 | 4.1 | 5.7 | 2.1 | 480 | 1.1 |
| 8/17/2017 | 17RF04 | 17 | 1.3 | 2.3 | 0.6 | 400 | 1.2 |
| 8/18/2017 | 17RF05 | 2.1 | 0.3 | 2.3 | 0.6 | 175 | 0.41 |
| 8/21/2017 | 17RF06 | 50 | 7.1 | 12 | 4.5 | 1250 | 4.2 |
| 8/24/2017 | 17RF07 | 18 | 2.1 | 4.2 | 1.5 | 1100 | 3.52 |
| 8/26/2017 | 17RF08 | 25 | 2 | 4 | 1 | 900 | 2.3 |
| 8/28/2017 | 17RF09 | | | | | 1600 | 6.1 |
| 8/30/2017 | 17RF10 | 30 | 4.5 | 4 | 4.5 | 1300 | 5.8 |
| 8/31/2017 | 17RF11 | 50 | 9 | 9 | 5 | 1300 | 5 |







**Table 2.** 2018 research flights conducted between 9/27/2018 – 10/23/2018. All concentrations and temperature values are taken at a 5 minute average when SAPs are observed within the BB plume. Some data values were not available and were left blank. Soot number and mass concentrations were obtained from the SP2. Organics, nitrates, sulfates, and ammonium were obtained from the AMS. Plume age was determined from HYSPLIT and WRF models.

| Date | Flight | Latitude | Longitude | Altitude (m) | Temperature (C) | # of SAPs | HYSPLIT Source Region | Plume Age (Days) |
|---|---|---|---|---|---|---|---|---|
| 9/27/2018 | 18RF01 | 11.3°S | 5°E | 1500 | 21.3 | 205 | Southern Congo | 4.5 |
| 9/30/2018 | 18RF02 | 7.85°S | 5°E | 1500 | 20.2 | 98 | Southern Congo | 5 |
| 10/2/2018 | 18RF03 | 7.75°S | 5.5°E | 2500 | 15.0 | 102 | Eastern Angola | 2 |
| 10/3/2018 | 18RF04 | 7°S | 6.3°E | 1300 | 19.4 | 48 | Northern Angola | 2 |
| 10/5/2018 | 18RF05 | 9.8°S | 6.2°E | 2100 | 16.7 | 39 | Central Angola | 2 |
| 10/7/2018 | 18RF06 | 10°S | 5°E | 2400 | 16.2 | 0 | Zambia | 4 |
| 10/10/2018 | 18RF07 | 13°S | 5°E | 2100 | 21.0 | 51 | Northern Namibia | 3 |
| 10/12/2018 | 18RF08 | 2°S | 6.6°E | 2000 | 17.1 | 0 | Northern Angola | 4 |
| 10/15/2018 | 18RF09 | 10°S | 5°E | 1600 | 22.5 | 98 | Southern Angola | 3 |
| 10/17/2018 | 18RF10 | 7.4°S | 10.5°E | 1900 | 20.0 | 16 | South America | 6 |
| 10/19/2018 | 18RF11 | 9°S | 7°E | 2500 | 16.2 | 35 | Southern Angola | 3 |
| 10/21/2018 | 18RF12 | 9°S | 5°E | 2100 | 17.3 | 0 | Tanzania | 5 |
| 10/23/2018 | 18RF13 | 1°S | 5°E | 2000 | 17.0 | 0 | Republic of Congo | 3 |

| Date | Flight | Organics Concentration ($\mu g\ m^{-3}$) | Nitrate Concentration ($\mu g\ m^{-3}$) | Sulfate Concentration ($\mu g\ m^{-3}$) | Ammonium Concentration ($\mu g\ m^{-3}$) | Soot Number Concentration (# $cm^{-3}$) | Soot Mass Concentration ($\mu g\ m^{-3}$) |
|---|---|---|---|---|---|---|---|
| 9/27/2018 | 18RF01 | 9 | 1.2 | 3.3 | 1.3 | 670 | 2.7 |
| 9/30/2018 | 18RF02 | | | | | 452 | 1.63 |
| 10/2/2018 | 18RF03 | 8.1 | 0.5 | 1.2 | 0.8 | 350 | 1.1 |
| 10/3/2018 | 18RF04 | 1.2 | 0.1 | 0.8 | 0.12 | 200 | 0.55 |
| 10/5/2018 | 18RF05 | 7.8 | 0.5 | 1.6 | 0.5 | 475 | 16 |
| 10/7/2018 | 18RF06 | 5.8 | 0.4 | 1.2 | 0.4 | 420 | 1.25 |
| 10/10/2018 | 18RF07 | 4.8 | 0.6 | 0.9 | 0.3 | 355 | 1.2 |
| 10/12/2018 | 18RF08 | 2.3 | 0.2 | 0.3 | 0.1 | 94 | 0.12 |
| 10/15/2018 | 18RF09 | 4 | 0.25 | 1.1 | 0.25 | 475 | 1.1 |
| 10/17/2018 | 18RF10 | 20 | 3.1 | 3 | 2.1 | 1100 | 2.5 |
| 10/19/2018 | 18RF11 | 3.7 | 0.2 | 0.6 | 0.1 | 275 | 0.52 |
| 10/21/2018 | 18RF12 | | | | | 175 | 0.55 |
| 10/23/2018 | 18RF13 | | | | | 176 | 0.59 |





Plume age was estimated from the backward trajectories run with HYSPLIT and ranged from 2-6 days (Tables 1 & 2). From HYSPLIT, backward trajectories were run until the air parcel was within 500 m of the surface. In 2018 the number of SAPs observed in relation to plume ranged widely. For example, 205 SAPs were measured on 27 September 2018 within an estimated 4.5 day old plume, whereas 102 SAPs were measured on 02 October 2018 within an estimated two day old plume. The complete set of backward trajectories from all of the research flights
conducted showed Angola was the most common source region in both 2017 and 2018 (Fig. 11).



**Figure 11:** Location of 2017 and 2018 aerosol sampling legs and their 48 hr HYSPLIT backward trajectories. Backward trajectories of SAPs (Blue) show a possible source airmass region in north and central Angola.

        The SAPs observed during 2017 and 2018 had maximum dimensions between 10 µm and 1520 µm and were linear or spherical in shape as example shown in Figs 2 & 8. In 2017, 165 SAPs were measured on six of thirteen
research flights while in 2018, 692 SAPs were measured on seven of the thirteen research flights (Tables 1 & 2). Analysis of the SAP sizes measured each year illustrate that the SAPs most commonly had D < 200 µm (Fig. 12). From 2017, five of the six research flights containing SAPs also contained aerosols observed with the CAS (Fig. 4). On flights within BB plumes where SAPs were not observed by the 2D-S, the CAS instrument also did not detect aerosols, which implies that aerosols within plumes were sized < 0.51 µm.

In 2017, organic concentrations trended lower for the days that did not record SAPs. Concentrations were measured between 2.1 - 25 µg m$^{-3}$ on days without SAPS, compared to 18 - 50 µg m$^{-3}$ on days with SAPs (Table 1). In addition, nitrates, sulfates, and ammonium concentrations were also lower for days with no recorded SAPs. This suggests that BB plumes with higher concentrations of organics, nitrates, sulfates, and ammonium were more likely to contain SAPs as well.





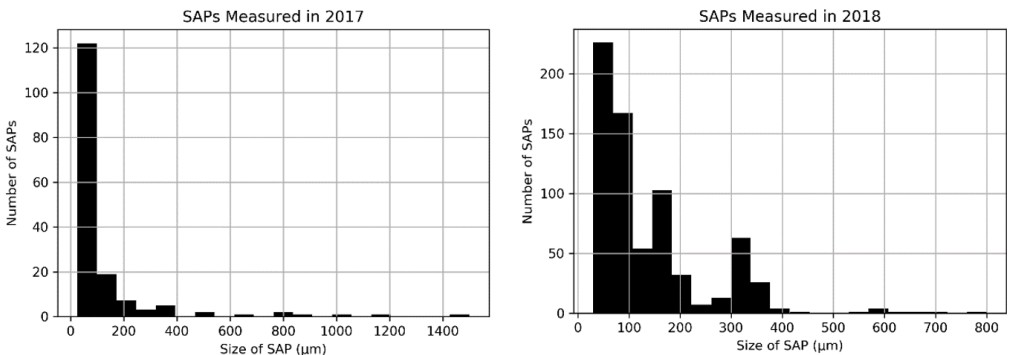

**Figure 12.** Particle sizes for the 165 SAPs measured in 2017 and the 692 measured in 2018.

In 2017 plume age was a major indicator of the presence of SAPs. Plumes aged between 2-3 days were the most likely to contain measurable SAPs. Flights flown through BB plumes aged between 3-8 days did not result in any SAPs observed on the 2D-S (Table 1). This suggests that either the SAPs were removed by gravitational settling, or not produced due to a difference in fuel type, fire intensity or wind speed and direction from the fire origin.

The maximum height of the aerosol plumes measured by CALIOP over South Central Africa during the months of August and October was on average 5500 m (Das et al., 2017). Assuming the SAPs were lofted to the top of the plume, their range of fall speeds can be estimated with the following bounds. The range of fall distance was estimated as the distance between the top of the plume and the top and bottom P-3 altitudes where SAPs were observed. The lowest altitude that SAPs were observed was 1230 m, and the highest was 3500 m. The bounds for the fall speed times were the longest and shortest times from the HYSPLIT back trajectories, which ranged from 48 hrs and 144 hrs. The range of fall velocities were then calculated to be at most, between 24.7 cm s$^{-1}$ and 0.4 cm s$^{-1}$.

In 2018 higher concentrations of organics (1.2 - 20 µg m$^{-3}$), as well as nitrate, sulfate, and ammonium, corresponded to times with measurable SAPs (Table 2). Out of the 10 research flights that recorded chemical composition consistent with BB, only two produced no observable SAPs on the 2D-S. There were too few days without SAPs during the 2018 campaign to determine whether chemical composition, plume age, or temperature was related to the presence of SAPs in the BB plume. The aerosol filter system provides the best insight concerning the composition of the BB plume and individual particles. The filter chemical analysis from 2017 show that the chemical composition of the particles was consistent with BB burning, specifically the EDS analysis showed peaks of K, and Si, both indicators of BB. In addition to the filter evidence, the HYSPLIT back trajectories showed a source location over areas of BB.

The main difference between 2017 and 2018 was the number of active fires present in south central Africa at the time of the flights. Fig. 5 illustrates the number of active fires present during the month of August 2017 and October 2018. In 2017 the entirety of south-central region of Africa around Angola, Congo, and Zambia had between 200-300 active fires of unknown intensity. This did not lead to more SAPs being sampled within the plume over the southeast Atlantic Ocean. In 2018 the BB season started to diminish, as is evident in the reduced frequency of active fires and the southward shift of the fires from Congo and Northern Angola. Nevertheless, more SAPs were observed. Due to differences in the locations of research flights between 2017 and 2018, it is possible that the smaller number of observed SAPs sampled in 2017 was related to sampling statistics rather than differences in SAP concentrations in the plumes.

**5 Conclusions**

This study examined characteristics of supermicron aerosol observed using in situ instruments on board the NASA P-3 Research Aircraft over the South Atlantic Ocean during the 2017 and 2018 NASA ORACLES field campaigns. Trajectory analyses using NOAAs HYSPLIT model, heightened concentrations of organics, nitrates, ammonium, and sulfates, a young plume age of 2-3 days, presence of refractory black carbon (rBC), and TEM-EDX-identified carbonaceous particles with enhanced K and Si peaks all support the hypothesis that the SAPs were associated with biomass burning within south central Africa. Similar particles emitted from biomass burning have



been observed previously in localized field studies as well as laboratory experiments, but not as far from the source as observed here. This work shows that SAPs can be transported hundreds of km from their source region. This analysis of SAPs focused on the shape, size, and composition of these particles based on optical array probe imagery, and analysis of aerosol filters. This study examined 15 research flights that recorded SAPs (out of 24) from 2017 and 2018.

In 2017, 165 SAPs sized 10 µm to 1520 µm were observed 820 km off the coast of Angola. The 2018 field campaign resulted in 692 SAPs 10 µm to 1100 µm observed 370-820 km off the coast of Angola. Filters collected during the 2017 campaign, containing particles sampled within the aerosol plume where the SAPs were observed, showed that collocated smaller particles were composed of black carbon and organics of biomass origin. NOAA HYSPLIT backward trajectories placed the source region of SAPs observed during 12 out of the 15 research flights in Angola for both 2017 and 2018. The SAPs composition was not measured. However, given the source location, presence of rBC, and the TEM-identified carbonaceous particles, it is hypothesized that based in observed particle shapes, the SAPs imaged by the 2D-S are examples of unburned plant material previously seen in biomass burning smoke. The exact impact of this class of large aerosols on aerosol-cloud interactions or cloud radiative processes is unknown, but maybe significant because of their apparent residence time in the atmosphere and their surface area. Future sampling of BB aerosol plumes hopefully will provide a better understanding of the lifecycle of SAPs and their potential role in radiative processes.

*Data Availability.* All ORACLES 2017 and 2018 in situ data used in this study are publicly available at https://doi.org/10.5067/Suborbital/ORACLES/P3/2017_V2 (ORACLES Science Team, 2020) and https://doi.org/10.5067/Suborbital/ORACLES/P3/2018_V2 (ORACLES Science Team, 2020). The ERA5 data (https://apps.ecmwf.int/data-catalogues/era5/?class=ea last access: 20 October 2020) are downloadable.

*Author Contributions.* RMM and GMM conceived the study design and analysis. JRO processed the in situ cloud probe data. RMM analyzed the data with inputs from GMM, RMR, SG, and JRO. RMM and GMM acquired funding. RMM, GMM, SG, JRO, MSR, AD, AJS, SB, SF and SH collected data on board the NASA P-3. CD analyzed filter samples. RMM wrote the paper with reviews from co-authors.

*Acknowledgements.* The authors wish to acknowledge the entire ORACLES science team, NASA Ames Earth Science Project Office and the NASA P-3 crew for the successful deployments. In addition we would like to thank Yohei Shinozuka for creating the merged instrument data files for all three years of the ORACLES campaign. We would also like to thank the NASA Ames Earth Science Project Office for their endless help and support throughout the missions.

*Financial support.* Funding for this project was obtained from NASA Award #80NSSC18K0222. ORACLES is funded by NASA Earth Venture Suborbital-2 grant NNH13ZDA001N-EVS2. RMM was supported by NASA headquarters under the NASA Future Investigators in NASA Earth and Space Sciences and Technology grant 80NSSC19K1371.





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
