# Peer review of "Observations of Supermicron-Sized Aerosols Originating from Biomass Burning in South Central Africa"

_Atmospheric Chemistry and Physics, 2021_

## Author Comment (AC1)

We thank the reviewers for their thoughtful comments. The paper has improved significantly as result of the changes they recommended.

**Response to reviewer 1:**

General comments:

This manuscript by Miller et al. observed supermicron aerosol particles over the Atlantic Ocean, ~1000 km from the coastal line. They conclude that these particles are unburned plant material originated from biomass burning. Their finding is interesting and could be important to understanding biomass burning and its regional influence. On the other hand, I found that this manuscript needs to be largely revised and improved to support the conclusion. I also found many technical issues to be revised.

**Major comments:**

1: The authors conclude that the supermicron aerosol particles (SAPs) are unburned plant materials (e.g., page 1 line 36). However, I could not find strong evidence to support that the SAPs are unburned plant materials. BC and other particles may not be excluded. In page 2 line 61-66, the authors showed the possibility of large BC. Burned plant materials (ash) may also be possible.

We agree with the reviewer that there is a possibility that the SAPs are large BC aerosol or ash. In the revision we have changed the sentences referring to unburned plant material to include these possibilities.

I could not find the reference that supports "previously seen in biomass burning smoke close to the source (page 1 line 37)". Here and relevant discussions (see below) are confusing and contradictive.

We rewrote the sentence in the abstract (where references are not permitted). At other locations in the text where appropriate, we have added references to the field study of Chakrabarty et al., 2014, and laboratory study of Kearney and Pierce, 2012.

Page 2 line 70-74: Evidence from transmission electron microscopy (TEM) analysis of aerosol particles, BB aerosol composition analysis, particle shape and size, and prevailing atmospheric conditions together demonstrates that SAPs observed during this campaign were not soot, but rather supermicronsized unburnt plant material.

This sentence was changed to: "Evidence from transmission electron microscopy (TEM) analysis of aerosol particles, BB aerosol composition analysis, particle shape and size, and prevailing atmospheric conditions together demonstrates that SAPs observed during this campaign were either black carbon aerosol, or supermicron-sized ash or unburned plant material."

Page9 line 246-248: The particles found on RF 12 most likely were either unburned plant material or supermicron black carbon aggregates that were formed near the fire and transported in the BB plume over the Atlantic Ocean.

This sentence was changed to: "The particles found on RF 12 most likely were either ash, unburned plant material or supermicron black carbon aggregates that were formed near the fire and transported in the BB plume over the Atlantic Ocean."

Page 9 line 254-256: The filters contained numerous black carbon and organic particles that were captured during the time that the 2D-S observed SAPs. It is therefore likely that the SAPs are unburnt plant material.

This sentence was changed to: "The filters contained numerous black carbon and organic particles that were captured during the time that the 2D-S observed SAPs. It is therefore likely that the SAPs are black carbon, ash or unburned plant material."

Page 19 line 402-405: However, given the source location, presence of rBC, and the TEM-identified carbonaceous particles, it is hypothesized that based in observed particle shapes, the SAPs imaged by the 2D-S are examples of unburned plant material previously seen in biomass burning smoke.

This sentence was changed to: "However, given the source location, presence of rBC, and the TEM-identified carbonaceous particles, it is hypothesized that based in observed particle shapes, the SAPs imaged by the 2D-S are examples of black carbon aerosol, ash or unburned plant material."

**Minor and technical comments:**

2. Page 1 line 22-23 (maximum dimension): What does the maximum dimension mean here? Is this Feret diameter?

The maximum dimension is the longest dimension in any direction across the 3D volume of the particle (D). We could not add this to the abstract because of word limits. Instead we added it to the text.

We added the following sentence and reference to the paper: "For all SAPs the longest dimension in any direction across the 3D volume of the particle (D) was recorded. D is equivalent to the diameter of the smallest sphere enclosing the particle (Wu and McFarquhar, 2016)."

3. Page 1 line 23 (supermicron-sized aerosol particles (SAPs)): "SAPs" is also defined as "supermicron aerosol particles (Page 2 line 67)".

All instances of supermicron were changed to supermicron-sized.

4. Page 1 line 27: black carbon (rBC): rBC is defined as refractory black carbon (page 3 line 139), and black carbon is defined as BC. Please make them consistent.

All references to black carbon measured by SP2 are now referred to as rBC. All other references to black carbon are referred to as BC. The abbreviations are now introduced at their first occurrence and used consistently throughout the paper.

5. Page 1 line 29: "black carbon": Here and elsewhere, black carbon is used as either "black carbon" or BC. Please make it consistent.

See comment above.

6. Page 1 line 28-31: "Transmission electron microscopy images of submicron particulates, collected on Holey carbon grid filters, revealed particles with potassium salts, black carbon and organics while energy-dispersive X-ray spectroscopy spectra detected potassium, a tracer for biomass burning, indicating that the submicron particles originated from biomass burning in addition to black carbon." This sentence is hard to interpret.

This sentence was split into three sentences for clarity.

"Transmission electron microscopy images of submicron particulates, collected on Holey carbon grid filters, revealed particles with potassium salts, black carbon (BC) and organics. Energy-dispersive X-ray spectroscopy spectra also detected potassium, a tracer for biomass burning. These measurements provided evidence that the submicron particles originated from biomass burning in addition to BC. "

7. Page 2 line 58: "KCL" should be KCl.

Change made.

8. Page 2 line 62: Please define "Dp". "D" is defined as "maximum dimension" in abstract. Is this the same?

Since D hasn't yet been defined in the paper the sentence was changed to read:

"Large soot aggregates ( >1000 nm) have been observed in field studies of flaming wildfires over the southern Indian Ocean and the Southwestern USA (Chakrabarty et al., 2014)."

The interested reader can check the reference for their definition of size.

9. Page3 line 97: "No SAPS were detected during the 2016 IOP." This sentence may be placed in the result section. "SAPS" should be SAPs.

We left this point were it was. However in response to reviewer 2, we added a discussion of reasons why SAPs were not observed in 2016. That discussion is in section 3.1.

10. Page 4 line 148: "The inlet size for the AFS allowed aerosols of 2.0 - 3.1 μm diameters" This sentence reads the measured sizes are between 2.0 and 3.1 μm. However, in the next paragraph (2.7), these sizes are 50% transmission efficiency of the inlet. It looks like something is missing.

The issue was resolved by reversing the order of sections 2.6 and 2.7 and deleting the sentence:

"The inlet size for the AFS allowed aerosols of 2.0 - 3.1 μm diameters, with a 50% collection efficiency depending on the altitude, to be sampled."

11. Page 5 line 200: "mb" may need to be replaced with hPa.

All instances of mb were changed to hPa.

12. Page 5: Figure 1: Flight tracks are difficult to read because of the color.

All flight tracks switched to yellow.

13. Page 6 Figure 2: I am not familiar with the instrument, but is there any artifact caused by aircraft speed for the long, straight particles on the 2D-S prove images?

No. The aircraft-imaging rate is synced to the true airspeed so the pixel dimension in particle images is constant. The probe is mounted ahead of the wing and is not influenced by the flow around the aircraft. This is routinely tested when pylons are developed for NASA aircraft.

14. Page 9 line 253-255. "The soot particles showed silicon inclusions and the organic particles contained potassium, both molecular markers for biomass burning emissions (Andreae et al. 1998)." I could not find that silicon inclusions are a marker for biomass burning emission in Andreas et al. (1998) or any other references. I think silicon can originate from various sources, including biomass burning, dust, and plant, and is difficult to be used for the biomass burning tracer.

We rewrote the sentence to read: "The soot particles showed carbon, sulfur, oxygen and silicon inclusions and the organic particles contained potassium, which are characteristic of biomass burning emissions (Pósfai et al., 2003).

15. Page 10 line 259. Here the sentence mentions that "gold grid" was used for the sampling. On the other hand, in Figure 7, I found Cu signal possibly from the grids but no Au signal. Please check it.

The grid was copper, our mistake.

16. Page 11 Figure 7. It is difficult to read the elements, some of which may be under the detection limits (e.g., Na, S, and Cl). It looks N has a large difference between round organic and soot.

We redrafted the figure to make everything readable.

17. Page 12 line 282. Add a period.

Period added.

18. Page 13 line 308-310: "Higher intensity fires with active flaming with high combustion temperatures most likely produce more SAPs than smaller smoldering fires as higher intensity fires inject more aerosols and unburned plant material into the atmosphere." It is possible that active flaming emits not only unburned plant materials but also various SAPs (e.g., BC).

The sentence was removed in response to reviewer 2.

19. Page 14 Figure 10. What do the colors indicate?

Colors indicate flights on different days. Explanation was added to the caption.

20. Page 15 Table 1. The decimal places are inconsistent. An example is soot mass concentration (1.75, 1.1, and 5).

The decimal places were all corrected so they use the same number of significant figures.

21. Page 18 Figure 12: The bin sizes are different in the two panels. Please consider using the same bin size for both panels so that both panels are directly comparable.

The bin sizes are all now 10 um.

22. Page 18 line 364-366: "The bounds for the fall speed times were the longest and shortest times from the HYSPLIT back trajectories, which ranged from 48 hrs and 144 hrs. The range of fall velocities were then calculated to be at most, between 24.7 cm s-1 and 0.4 cm s-1." This estimation could be interesting. I suggest considering if the estimated fall velocities are reasonable for the measured SAPs by assuming their sizes and densities.

We explored this issue quite a bit when writing the original paper. The problem we have is that the density and porosity of the SAPs are unknown. We were unable to find any references in the existing literature to put bounds on these parameters. As a result, any attempt we make to determine whether the fall velocities are reasonable would be guesswork and totally speculative. For this reason we did not make any changes to the paper.

23. Page 18 line 369-371. "There were too few days without SAPs during the 2018 campaign to determine whether chemical composition, plume age, or temperature was related to the presence of SAPs in the BB plume." This sentence seems strange. Does it mean "too few days with SAPs"? If not, it may be discussed using airmasses with SAPs.

Correct. It should have been with SAPs. Change made.

24. Page 18 line 391-and Page 19 404-405 "Similar particles emitted from biomass burning have been observed previously in localized field studies as well as laboratory experiments" and "the SAPs imaged by the 2D-S are examples of unburned plant material previously seen in biomass burning smoke" Where do these sentences indicate in the main text (result or discussion)? I could not find the previous study in the reference.

The first sentence in the conclusion was revised to include references. These references also now appear in the introduction. The sentence in the conclusion now reads:

"Similar particles emitted from biomass burning have been observed previously in localized field studies (Chakrabarty et al., 2014) as well as laboratory experiments (Kearney and Pierce, 2012), but not as far from the source as observed here."

The second sentence was revised to read:

"However, given the source location, presence of rBC, and the TEM-identified carbonaceous particles, it is hypothesized that based in observed particle shapes, the SAPs imaged by the 2D-S are examples of supermicron-sized BC aerosol, ash or unburned plant material."

25. Page 20 References: The reference order is not consistent. An example is those on page 20 line 466-474.

References were reordered.

Suggestion: Many references were published more than 20 years ago, and some can be replaced with recent ones. An example is in page 1 Line 41: "Particulates generated by BB scatter and absorb solar radiation, affect the properties and lifetime of clouds (Andreae 1991, Penner et al. 1992, Ackerman et al. 2000, Bond et al. 2013), and influence regional and global climate (Crutzen & Andreae 1990, Andreae 1991; Bond et al., 2013)."

It was unclear which references the reviewer wanted us to add since none where provided. In referencing we chose to reference seminal papers on the subject. We will be happy to add other references if the reviewer can direct to specific references that they believe are missing.

**Response to reviewer 2:**

This work reports observations of supermicron-sized aerosol particles (SAPs) using a wing-mounted imaging probe during the ORACLES airborne measurement campaign. Coincident measurements of aerosol composition and corroborating back trajectories support a biomass burning source for these SAPs. The authors hypothesize that the SAPs are unburned vegetative matter (i.e., grass) that is convectively lifted and transported by fires in south-central Africa.

**Major Critiques**

The number of individual SAP particles identified during extensive smoke-plume sampling is extremely low, and the authors opt to not even calculate their concentrations and rather simply report counts. Thus, it feels that the paper lacks sufficient 1) presentation of the actual data, 2) second-level analysis of SAP properties, and 3) justification for the vegetative source.

1. With so few of the SAP particles identified, it would be very informative to show more 2DS images in the paper. My suggestion would be to show a full-page figure of all SAPs for each of the case studies. This will allow readers to get a better sense of the variation in particle shape and would help to corroborate the unburned-grass hypothesis.

   Thank you for the suggestion. We have replaced the original figures showing the SAPs within each case study with figures 3 and 10 that show many more SAPs organized based on shape and size. We added text regarding the shapes of the SAPs in both case studies. We are unable to show all SAPs (there are 175 in one case study), but now show a representative sample that characterizes the full dataset. We also show a size distribution in figure 14 categorized by shape

Second, the paper is focused solely on observations in the African smoke plume, but are there any SAPs that were identified outside of the plume?

There were no SAPs identified outside the plume. The aircraft rarely sampled outside of the plume during the project. We added a sentence stating that all SAPs discussed in the paper were detected within the BB plumes.

A comparison of the BB data with data in background boundary layer or for free-tropospheric conditions would strengthen the argument that the SAPs are truly being emitted with smoke.

By "background boundary layer" we assume the reviewer means the boundary layer outside of the BB plume. During ORACLES the aircraft never sampled the boundary layer in conditions outside the plume, so we have no aircraft data to address this component of the comment. The aircraft did sample the free-troposphere above the aerosol plume. No SAPs were observed in this region on any flight. We have added a sentence stating this to the paper and we have added the time series plots that reviewer requested which shows this specifically.

2. There is some qualitative discussion of the SAP particle shape in the text, but a more systematic and quantitative analysis strengthen the paper. First, it is not clear how many of the SAPs are identified in only one 10-um pixel and whether those single pixel detections can be confidently counted as real particles and not a sampling artifact biasing the results.

We have added the systematic and quantitative analysis requested by the reviewer in the new figure 14. This figure shows the complete size distribution of all SAPs measured during ORACLES as well as their distribution related to their shapes. To address this comment we have formatted the figure so that the bins are at the resolution of the probe. The number of single pixel SAPs now is obvious in the first bin. We are convinced that these are not artifacts because relative to the total time of the flight outside of cloud these images rarely occurred, but when they did they were near other SAPs. These never occurred in the free troposphere above the plume. If the probe electronics generated random artifacts we should see these both above and within the plume. For the 2D-S probe the diode voltage has to drop 50% across a bit to record a particle. This is to ensure false counts are not made.

The number of single-pixel counts need to be explicitly presented in the text.

The number of single-pixel counts now appear explicitly in figure 14.

Second, can the SAPs be grouped into similar shapes to quantify those that are grass-like (i.e., rods, elongated in one dimension) compared to other more spherical shapes? With so few particles, this grouping could be done manually and likely would not necessitate a mathematical clustering approach similar to cloud morphology studies. Some form of shape clustering would be helpful, especially to justify the grass source theory.

All SAP images are now categorized by shape and the summary statistics are presented in figure 14.

3.  The source of the SAPs is purely speculative and needs some further justification. Shape analysis (above) would help, but at least providing a more rigorous literature feasibility study is necessary. Has vegetative material been identified using cloud imaging probes before? Are there images of these particles in the literature that could be compared to the measurements presented here?

    In figure 14 we now provide a shape analysis. Despite several extensive searches, we have not found any literature corresponding to plant material being measured by cloud imaging probes. Many of our coauthors are experts on BB plumes and none are aware of existing literature regarding vegetative material being imaged by imaging probes on aircraft. Virtually all the work done on BB plumes involving aerosol filter analysis, which is dominated by sub-micron particles.

    Can any of the variability in counts flight-to-flight be related to the presence or type of vegetation in the source area?

    We do not have data necessary to answer this question. The type of vegetation in central Africa varies from field to field, and back trajectory calculations have insufficient resolution to track back to individual fields. In addition during transport to the aircraft observation point, mixing undoubtedly occurs between plumes originating from different vegetation sources making it impossible to track back to specific vegetation.

Minor Edits

Page-Line

2-50      Please clarify this sentence.  I'm not sure what you mean by "still contributes to uncertainty estimates of RF".

We agree that the word "still" is confusing. It was removed from the sentence.

2-56      A more recent reference is Shingler et al. [2016] for soot restructuring (https://agupubs.onlinelibrary.wiley.com/doi/full/10.1002/2015JD024498).

Reference of Shingler et al. was added to the paper.

2-58      Please provide a reference for the KCl age statement.  Also this should be "KCl" not "KCL".

 The reference we provided to Li et al., 2003 applies the both sentences in the paragraph, and address the KCl age statement. "KCL" was changed to KCl.

2-76      Can you further comment here, or more appropriately in the discussion section, about why no SAPs were identified in 2016?

A paragraph was added in section 3.1 where we first refer to figure 5 discussing possible reasons why SAPs were not observed during the 2016 campaign. Our best interpretation is that the 2016 campaign was out of Walvis Bay, Namibia, where, as figure 5 and the figure below shows, the area in Africa is a bleak desert, while the area around Gabon (near Sao Tome) is a rainforest. The great circle distance between Sao Tome and Walvis Bay is 2,721 km (red line). The Walvis Bay site was initially chosen because of the consistency of the stratiform cloud deck over the Eastern Atlantic. Namibia would not allow flights in IOP years two and three, which necessitated the move to Sao Tome. The sampling areas downwind of Africa are obviously quite different as can be seen from the map. We added boxes on figure 5 showing the region of the flights in 2016 vs 2017-2018 and refer to that in the new paragraph.

[Figure]

2-85    change "difference" to "differences"

Word changed.

3-104   Can you comment in the 2DS section about instrument noise?  What is the false-count rate in clear air? Given that the number of SAPs is so low, I think it's worth the effort to quantify the false-count rate in order to trust the SAP observations. Also for this section, can you include a sentence or two on calibration of the probe?

Information about instrument noise and calibration of the probe was added to section 2.2. Calibration of the instruments, which includes laser alignment and sizing, was performed by the manufacturer prior to and after each experiment. Instrument maintenance was performed within the field to ensure cleaned optics before each flight. We are confident that the larger SAPs are real because the particles have a canting angle (angle off of the direction of flight, usually 15-30 degrees, due to flow distortion around the probe head) within the images, which is a firm indicator that they have mass and are not false-counts. This is a well known phenomena in 2D-S images. We also examined the 2D-S data when the aircraft was above the plume and in low concentrations of aerosol and found no false counts on the 2D-S giving us more confidence that the counts associated with the smaller SAPs are real. This is now noted in the paper.

3-126   The AMS vaporizer is typically operated at a nominal 600C temperature, see the cited DeCarlo [2006] reference.  Can you comment on why it was run at 650C for these flights and if that affects the AMS data presented?

The sentence was inaccurate and should have said from 600°C - 650°C. On a level leg in the plume during the first IOP in 2016, the instrument PIs set the temperature of the vaporizer to 500˚C, 600˚C,

and 700˚C to observe how the concentration of real time organic aerosol changed. The nephelometers and long differential mobility analyzer (LDMA) showed constant aerosol concentrations. The measured concentration peaked at 600˚C indicating that 600˚C was optimal for this aerosol. Temperatures below 600˚C did not efficiently sample organic aerosol, meaning the heated surface was not hot enough to evaporate the organic aerosol. Above ~700˚C was too hot, and we believe some of the organic aerosol then begins to char on the heated surface and leave a sticky burnt residue behind. Temperatures between 600˚C and 650˚C were used for the entire campaign.

3-129    Since you state that the AMS measurements are quantitative, please include a statement about the collection efficiency that was used to calculate final mass concentrations.

96% of the samples had a collection efficiency of 0.5, which has been regarded as close to accurate except when NH4+:SO4= ratios are low. No allowances were made for sampling delay due to inlet tubing length. A default fragmentation pattern was assumed. We added the following sentence "The applied collection efficiency, CE=max(0.5, 1- $NH_4$/(2x$SO_4$), neglects the small nitrate contribution, and used 0.5 as the lower limit, consistent with most field campaigns."

4-177    I don't think the Eloranta [2008] reference is appropriate for the HSRL-2 instrument. I suggest Hair et al. [2008] for a general reference for the instrument (https://www.osapublishing.org/ao/fulltext.cfm?uri=ao-47-36-6734&id=175351) or just use Burton et al. [2018] (https://www.osapublishing.org/ao/fulltext.cfm?uri=ao-57-21-6061&id=395340)

We used Burton et al., 2018.

5-199    For this section, I highly suggest adding time-series plots for the case study. This should show, at a minimum, altitude, 2DS counts for SAPs, LWC to confirm a lack of cloud, CO and BC to illustrate the location of the smoke plume.

Time series plots were added for each case study. The time series plots show the entire flight  with all the variables that the reviewer requested plus the locations of the SAPs, the stratiform cloud top height range, and the aerosol plume top height range. Paragraphs were added to describe these figures.

Also, please comment on the following: Are the SAP detections randomly observed or do they cluster in time?

The time series plots we added show that the SAPs were clustered in several altitudes and locations.

Are there SAP detections outside of the smoke plume?

All SAP observations with found within the smoke plume.

Are the SAPs more frequently observed at a specific altitude?

As show on figures 2 and 9 SAPs occur at many altitudes.

As stated above, I recommend that all 72 + 12 SAP images should be shown and discussed for this section.

See response to major comment.

6-209    The text states that RF11 had 71 SAPs measured but TABLE-1 shows only 12. TABLE-1 also does not have RF12. Please check the text and table and confirm this discrepancy.

We mislabeled the RFs in the table. Good catch!

8-230    I do not understand the following sentence, "The CAS did not report…". Please clarify.

We replaced the sentence with the following sentence. "The CAS did not report any counts in clear air above the aerosol plume."

8-232    What is the conclusion drawn from the CAS data? There seems to be two orders of magnitude difference in concentrations of particles > 10um diameter for RF11 and RF1, but these flights have nearly the same number of SAPs observed (12 and 15 SAPs, respectively, from Table-1). Please comment on this discrepancy. Also, why were these 5 flights chosen for the plot, and what about 2018 flights? In general, I suggest inclusion of a more succinct description of the utility of the CAS data for this plot to be useful and included in the final paper.

This is an excellent point and caused us to rethink why the concentrations would be different between the probes. The CAS operates as a forward scattering probe while the 2D-S imaging is based on occultation of a photo diode array based on 50% occultation. The fact that the CAS is seeing higher concentrations strongly suggests that the 2DS is missing many of the particles near 10 µm because they are not sufficiently occulting the photo diode array. We have added text to the paper addressing this issue. We noted in the instrumentation section that the CAS was not operational for the entirety of the 2018 campaign.

9-256    Are the soot, organic, and dust particles that you reference here shown in Figure-6?

Yes. Most likely these particles contain all soot, organic, and dust.

9-255    "It is therefore likely that the SAPs are unburnt plant material." I do not follow this argument. Please provide justification.

We changed the language to "It is therefore likely that the SAPs are examples of supermicron-sized BC aerosol, ash, or unburned plant material." This was based on a recommendation of another reviewer.

11-263   As with the first case study, I highly suggest addition of a time series to address the questions posed above. Even though there are more SAP observations, showing each image would be beneficial.

Time series were added.

12-278   "SAPs could be correlated with fire intensity".  This statement is very speculative and needs to be justified or removed. I do not know what "copious amount" of material means, please explain.

The sentence was changed to "The production of SAPs may be related to fire intensity as very intense fires can inject more material into the atmosphere as seen in SAFARI-92 (Canut et al., 1996)."

13-292   "… were always observed within the BB aerosol plume…" As written, there is no evidence for this statement.  Time series of the case studies showing both BB and non-BB sampling would be helpful. Statistics for BB and non-BB sampling would also help.

Time series plots for the case studies are now presented and were examined for all flights to confirm this statement. We stand by this statement after this analysis.

13-303   "This could have resulted from an increase in fire intensity."  This statement is very speculative and needs to be justified or removed.

The two sentences related to speculation regarding fire intensity were removed.

14-322   "Aerosol plumes where SAPs were observed…".  I do not understand this statement.  Why is the "spread" in the AMS data indicative of more intense fires?  Please explain.

We deleted the phrase "which would be consistent with more intense fires producing more emissions." And added a reference to Table 1 to the sentence.

15-325   The caption states that data are shown for "5 minute average when SAPs are observed within the BB plume", but for many of the flights the SAP count was zero and data are still shown?  I'm confused.  Please clarify.

The sentence was clarified  to read "All concentrations and temperature values are 5 minute averages."

Also, RF5 (2017) has more sulfate than organics and therefore does not appear to be from BB.  Please comment. Same argument for RF4 (2018).

This may be true for RF5 (2017). There were no SAPs on that flight. RF4 (2018) had low concentrations of both organics and sulfates but there were more organics than sulfates based on the data in the table.

18-355   What is the bin width of the histograms?  How many of the counts are for 10um particles (i.e., for a single pixel).  Please comment on if you think these are real or a possible artifact.

The bin width is now stated in the text (10 µm). The figure was remade at the 10 µm resolution of the 2D-S. We added information concerning real vs. artifact images, see previous comment.

---

## Author Response (AR2)

Response to reviewer 1

Page 1 line 31. "biomass burning.." Two periods here.

One period was removed.

2. Page 3 line 92. "(Redemann et al 2020)" A comma needs after "et al".

A comma was added to the reference.

3. Page 8. Figure 3. What are the blue dot lines and black straight lines?

Blue dot lines and black straight lines were boundaries of the images. They should not have appeared and we removed them.